# Novel Multifunctional Luminescent Electrospun Fluorescent Nanofiber Chemosensor-Filters and Their Versatile Sensing of pH, Temperature, and Metal Ions

**DOI:** 10.3390/polym10111259

**Published:** 2018-11-13

**Authors:** Bo-Yu Chen, Yen-Chen Lung, Chi-Ching Kuo, Fang-Cheng Liang, Tien-Liang Tsai, Dai-Hua Jiang, Toshifumi Satoh, Ru-Jong Jeng

**Affiliations:** 1Institute of Polymer Science and Engineering, National Taiwan University, 106 Taipei, Taiwan; leodapon@gmail.com; 2Institute of Organic and Polymeric Materials, National Taipei University of Technology, 10608 Taipei, Taiwan; iduo1125@hotmail.com (Y.-C.L.); frank62112003@yahoo.com.tw (F.-C.L.); x1x2x310@gmail.com (T.-L.T.); asdf8289@yahoo.com.tw (D.-H.J.); 3Research and Development Center for Smart Textile Technology, National Taipei University of Technology, 10608 Taipei, Taiwan; 4Graduate School of Chemical Sciences and Engineering and Faculty of Engineering, Hokkaido University, Sapporo 060-8628, Japan; satoh@eng.hokudai.ac.jp; 5Advanced Research Center for Green Materials Science and Technology, National Taiwan University, 106 Taipei, Taiwan

**Keywords:** electrospun nanofibers, pH, mercury ion, chemosensor, thermo-response

## Abstract

Novel multifunctional fluorescent chemosensors composed of electrospun (ES) nanofibers with high sensitivity toward pH, mercury ions (Hg^2+^), and temperature were prepared from poly(*N*-Isopropylacrylamide-*co*-*N*-methylolacrylamide-*co*-rhodamine derivative) (poly(NIPAAm-*co*-NMA-*co*-RhBN2AM)) by employing an electrospinning process. NIPAAm and NMA moieties provide hydrophilic and thermo-responsive properties (absorption of Hg^2+^ in aqueous solutions), and chemical cross-linking sites (stabilization of the fibrous structure in aqueous solutions), respectively. The fluorescent probe, RhBN2AM is highly sensitive toward pH and Hg^2+^. The synthesis of poly(NIPAAm-*co*-NMA-*co*-RhBN2AM) with different compositions was carried on via free-radical polymerization. ES nanofibers prepared from sensory copolymers with a 71.1:28.4:0.5 NIPAAm:NMA:RhBN2AM ratio (**P3** ES nanofibers) exhibited significant color change from non-fluorescent to red fluorescence while sensing pH (the λ_PL, max_ exhibited a 4.8-fold enhancement) or Hg^2+^ (at a constant Hg^2+^ concentration (10^−3^ M), the λ_PL, max_ of **P3-fibers** exhibited 4.7-fold enhancement), and high reversibility of on/off switchable fluorescence emission at least five times when Hg^2+^ and ethylenediaminetetraacetic acid (EDTA) were sequentially added. The **P3** ES nanofibrous membranes had a higher surface-to-volume ratio to enhance their performance than did the corresponding thin films. In addition, the fluorescence emission of **P3** ES nanofibrous membranes exhibited second enhancement above the lower critical solution temperature. Thus, the ES nanofibrous membranes prepared from **P3** with on/off switchable capacity and thermo-responsive characteristics can be used as a multifunctional sensory device for specific heavy transition metal (HTM) in aqueous solutions.

## 1. Introduction

Some studies have demonstrated the adverse effects of exposure to heavy transition metal (HTM) cations on children’s health, such as low birth weight, low anogenital distance, and severe DNA and chromosome damage. Exposure to methylmercury can lead to Minamata disease, the typical symptoms of which are sensory disturbances (glove and stocking type), ataxia, dysarthria, constriction of the visual field, auditory disturbances, and tremors in some cases [1,2,3]. Therefore, selective and sensitive chromogenic or fluorogenic chemosensors for the detection of various HTM cations [4,5,6] have been designed to solve the aforementioned issues. Kaewtong et al. investigated fluorescent chemosensors composed of a rhodamine-based scaffold and pseudo azacrown cation-binding subunit. The chemosensors exhibited on/off fluorescence emission toward mercury ion (Hg^2+^) chelation in an organic solution with a short response time, low background interference, low detection limits, long wavelength emission, and high selectivity [5]. Recent studies have demonstrated the high potential of rhodamine B (RhB)-copolymer-based chemosensors for the sensing of temperature, pH, and certain HTM cations in aqueous solutions because of their water solubility, simplicity, ability to change color, and high selectivity [7,8,9,10]. Liu et al. reported multifunctional fluorescent copolymers containing RhB moieties that could be used as ultrasensitive fluorometric chemosensors in aqueous media. Their multi-color emission can be effectively tuned by adjusting the temperature, pH [8], Zn^2+^ concentration [10], and Hg^2+^ concentration [7]. However, the primary concern with solution-based chemosensors is that they are not reusable. Wu et al. reported a reusable polymer film chemosensor prepared from a surface modified with a polyvinyl alcohol film probe that could mitigate this defect [11]. However, the aforementioned studies have all used solutions [7,8,9,10] or polymeric films [11,12] rather than ES nanofibers. Nanofibers with a high surface-to-volume ratio are expected to enhance responses for temperature, Hg^2+^, and pH sensing, and also provide reversibility.

The electrospinning process is inexpensive, easy to use, and versatile for producing sub-micron scale fibers, which has been used extensively in many studies and applications [13,14,15,16,17,18,19,20,21,22], such as coaxial electrospinning [23,24], side-by-side electrospinning [25], and other types of multiple-fluid electrospinning [26,27]. Various florescent electrospun (ES) polymer nanofiber-based chemosensors in our previous studies were designed for sensing pH [28,29,30,31,32], temperature [33,34,35], and NO gas [36]. Notably, some studies have reported fluorescent ES nanofiber-based chemosensors for sensing specific HTMs such as Fe^3+^ [29], Hg^2+^ [28,31,32,35,37], Zn^2+^ [30,37], and Cu^2+^ [31,37]. We previously reported a novel fluorescent chemosensory ES nanofibrous membrane with high selectivity and reversibility toward pH- and Hg^2+^, which was prepared using poly(hydroxyethyl-*co*-*N*-methylolacrylamide-*co*-rhodamine derivative) copolymers. In addition, the high surface-to-volume ratio for ES nanofibers affords them with higher sensitivity relative to that of thin films [28]. Moreover, ES nanofibers with thermo-responsive properties have attracted considerable attention because their morphology and diameter can be effectively tuned using environmental temperature, and thus they can be used in many applications such as new types of chemosensors and drug carriers [38,39,40]. Chiu et al. reported that cross-linked poly(*N*-Isopropylacrylamide-*co*-NMA) (poly(NIPAAm-*co*-NMA)) copolymer-based ES nanofibers could maintain their shape when immersed in water, during which time they exhibited different swelling performances, lower critical solution temperatures (LCST), and morphologies at various temperatures [39,40]. In the present study, we attempted to put together the aforementioned advantages to develop multifunctional ES fibrous chemosensors capable of temperature, pH, and Hg^2+^ sensing. The chemosensors prepared from sensory copolymers with different NMA ratios for cross-linking reactions exhibited different morphologies, LCSTs, and degrees of swelling when immersed in water. Most notably, the chemosensor can be used as a multifunctional device for pH-sensing and specific HTMs-chelating, and the intensity of fluorescence emission can be modulated by temperature tuning.

We report chromo- and fluorogenic sensory nanofibrous membranes prepared using poly(*N*-isopropylacrylamide-*co-N*-methylolacrylamide-*co*-rhodamine derivative) (poly(NIPAAm-*co*-NMA-*co*-RhBN2AM)) copolymers. Due to the presence of a hydrophilic and thermo-responsive moiety (PNIPAAm), a chemical cross-linkable moiety (PNMA), and a fluorescent probe with Hg^2+^-chelating and pH-sensing characteristics (RhBN2AM), the sensory nanofibrous membranes exhibited multifunctional sensory characteristics for Hg^2+^, pH, and temperature sensing. The nanofibrous membranes were prepared from poly(NIPAAm-*co*-NMA-*co*-RhBN2AM), which was synthesized through free-radical polymerization. The polymerization schematic and the sensing mechanism of RhBN2AM are shown in Scheme 1a. Their LCSTs, optical properties, and morphological characteristics were analyzed. The electrospinning process was used to fabricate ES nanofibers from poly(NIPAAm-*co*-NMA-*co*-RhBN2AM) (Scheme 1b), and the stability enhancement of nanofibers in water was achieved through chemical cross-linking reactions. In addition, Scheme 1c shows that the sensory ES nanofibers exhibited reversible thermo-induced second fluorescence intensity enhancement when the sample was heated above LCST and subsequently cooled down. The fluorescence emission of RhBN2AM is highly sensitive toward Hg^2+^ or pH. In the detection of Hg^2+^ or acidic media, RhBN2AM transformed from its spirocyclic form (colorless and non-fluorescent) into its opened cyclic form (pink and strong fluorescence emission) [5,28]. Thus, the on/off switching of the fluorescence can be easily modulated by tuning pH levels or Hg^2+^ concentration, whereas the fluorescence intensity can be easily tuned through heating or cooling. We further compared the fluorescence emissions of chemosensors prepared from thin films and ES nanofibrous membranes. Furthermore, the fluorescence emission of the novel chemosensors exhibited second enhancement above the LCST, which indicated that the chemosensors could be applied in pH (or Hg^2+^) and temperature simultaneously. The favorable reversible property of Hg^2+^ or pH sensing indicates that sensory nanofibrous membranes certainly possess the potential for applications in multisensory devices and HTM filtration.

## 2. Experimental Section

### 2.1. Materials

*N*-Isopropylacrylamide (NIPAAm), *N*-methylolacrylamide (NMA), and 2,2’-azobis(2-methylpropionitrile) (AIBN), were purchased from Sigma-Aldrich (St. Louis, MO, USA), Tokyo Chemical Industry Co. (Tokyo, Japan), and UniRegion Bio-Tech (Yesan-Gun, Korea), respectively. They were re-crystallized and stored at 4 °C prior to use. Trizma^®^ base (Sigma, St. Louis, MO, USA, 99.9%), Rhodamine base (RhB, Sigma-Aldrich, 97%), anhydrous magnesium sulfate (MgSO_4_, J.T. Baker, Phillipsburg, NJ, USA, assay), Ethylenediamine (Sigma-Aldrich, 99%), acryloyl chloride (Sigma-Aldrich, 97%), and triethylamine (TEA, Sigma-Aldrich, 99.5%) were used as received. Methanol (Tedia, Fairfield, OH, USA, HPLC/SPECTRO), ethanol (Sigma-Aldrich, absolute), acetonitrile anhydrous (Tedia, 99%), tetrahydrofuran (THF, Tedia, HPLC), and dichloromethane (Tedia, 99.9%) were used as received. All perchlorate salts of metal ions were obtained from Aldrich (Milwaukee, WI, USA).

### 2.2. Characterization

^1^H nuclear magnetic resonance (NMR) data were recorded at room temperature on a Bruker AM 300 (300 MHz, Billerica, MA, USA) spectrometer using the residual proton resonance of the deuterated chloroform and deuterated dimethyl sulfoxide. The LCST (phase transition) of the prepared copolymer solution was recorded by monitoring the transmittance of a 540 nm light beam on a Shimadzu UV–Vis spectrophotometer (Kyoto, Japan). The copolymer concentration in water was 1 wt %, and the temperature was raised from 25 to 70 °C in 2.5 °C increments for every 10 min [35,36]. Thermal analysis was carried out via a thermogravimetric analyzer (TGA) from TA instrument (TA Q500, New Castle, DE, USA) with a heating range from 100 to 700 °C at a heating rate of 10 °C min^−1^. The morphologies of ES nanofibers were investigated using a field-emission scanning electron microscope (SEM, Hitachi S-4800, Tokyo, Japan) and a two-photon laser confocal microscope (Leica LCS SP5, Wetzlar, Hessen, Germany). Photoluminescence (PL) spectra were measured to study photophysical properties, which were recorded using a Fluorolog-3 spectrofluorometer (Horiba Jobin Yvon, Kyoto, Japan). All PL spectra of ES nanofibers for pH and Hg^2+^ sensing were recorded using the Fluorolog-3 spectrofluorometer and excited at a wavelength of 540 nm, as described in the Appendix A, which is similar to the process followed in our previous studies [28,29,30,31,32,33,34,35,36,37].

### 2.3. Synthesis of the Fluorescent Probe (RhBN2) and Fluorescent Monomer (RhBN2AM)

The synthetic scheme for RhBN2AM is shown in Scheme 2a. The syntheses of fluorescent probes, RhBN2, and RhBN2AM, were performed according to the literature [5,28]. The detailed procedures are shown in the Appendix A. The chemical structure of RhBN2AM was characterized using ^1^H NMR (Appendix A) and ESI-MS (Appendix A).

### 2.4. Synthesis of poly(NIPAAm-co-NMA-co-RhBN2AM) 

The synthetic scheme for poly(NIPAAm-co-NMA-co-RhBN2AM) is shown in Scheme 2b. Poly(NIPAAm-*co*-NMA-*co*-RhBN2AM) copolymers with different NMA ratios were denoted as **P1**, **P2**, and **P3**, as listed in Table 1. In a round-bottomed two-necked flask, NIPAAm (1.59 g, 14.1 mmol), NMA (712.1 mg, 7.05 mmol), RhBN2AM (75.6 mg, 0.14 mmol), and AIBN (4.3mg, 0.026 mmol) were dissolved in ethanol. Nitrogen bubbling was maintained for 30 min to remove oxygen. Subsequently, the mixture was immersed in an oil bath at 70 °C for 24 h. The polymerization was quenched by exposure to air. In order to remove unreacted monomers, the solution was diluted with ethanol, precipitated in ether, and then freeze-dried until constant weight, and the polymer **P3** was subsequently obtained. Characterization of **P3** by GPC and ^1^H NMR are shown in Appendix A, respectively. The copolymer composition of **P3** based on the NMR spectrum is estimated to be 71.1:28.4:0.5. Number average molecular weight (*M_n_*) = 35,500 g mol^−1^ and polydispersity index (M*_w_*/M*_n_*) = 1.88 were obtained from GPC measurement using DMF as an eluent (Appendix A). ^1^H NMR (300 MHz, DMSO, δ, ppm; Appendix A): 8.08 (h), 7.25 (g), 5.46 (f), 4.43 (e), 3.79 (d), 1.88 (c), 1.43 (b), 0.93 (a).

### 2.5. Preparation of Electrospun Fibers and Drop-Cast Films

The ES nanofibers were fabricated using the electrospinning process as described in Scheme 1b, similar to the one used in our previous reports [28,29,30,31,32,33,34,35,36,37]. The polymer solutions of **P1**, **P2**, and **P3**, (400 mg mL^−1^ dissolved in a methanol (MeOH)) were filled into a metallic needle by using syringe pumps (KD Scientific Model 100, Holliston, MA, USA), with a feed rate set at 1 mL h^−1^. A high-voltage power supply (Chargemaster CH30P SIMCO, Capability Green, Luton, UK) was set at 10 kV during the ES process. On the other hand, the corresponding films of **P1**, **P2**, and **P3** were respectively drop-cast on glass substrates using the same concentration and volume as the polymer solution. For chemical cross-linking, all samples were treated at 120 °C for 48 h in an oven, which was similar to the process reported in our previous studies [28,29,31,35,36,37].

## 3. Results and Discussion

### 3.1. Characterization of RhBN2AM and poly(NIPAAm-co-NMA-co-RhBN2AM)

The chemical structure of RhBN2AM was characterized using ^1^H NMR (Appendix A) and ESI-MS (Appendix A). The synthetic routes of RhBN2AM are shown in Scheme 2a, which are similar to the ones in previous studies [5,28]. Poly(NIPAAm-*co*-NMA-*co*-RhBN2AM) copolymers with three different molar ratios of NMA, denoted as **P1**, **P2**, and **P3**, were synthesized via free-radical polymerization according to Scheme 2b. Appendix A shows the ^1^H NMR spectra of poly(NIPAAm-*co*-NMA-*co*-RhBN2AM) copolymers, which were composed of NIPAAm, NMA, and RhBN2AM in ratios of 90.2:9.4:0.4 (**P1**), 80.5:19.2:0.3 (**P2**), and 71.1:28.4:0.5 (**P3**) in DMSO. The proton peak for DMSO is shown in Appendix A. The proton peaks at 3.79 ppm (peak d) represent the methine neighbor of the methyl group on NIPAAm. The peaks at 8.08 (peak h) and 5.46 (peak f) ppm represent the secondary amine moiety and terminal hydroxyl moiety of NMA, respectively. Peak integrations for peaks g, h, and f increased with increasing NMA content. The peak at 0.93 ppm (peak a) represents the alkyl chains on the polymer. The copolymer composition estimated by performing peak integration was consistent with the proposed structure. The *M_n_*_, GPC (DMF)_ and *M_w_*/*M_n_* of the **P1**, **P2**, and **P3** copolymers were 33,000 and 1.85, 35,300 and 1.92, and 35,500 and 1.88, respectively, as shown in Table 1 and Appendix A. The LCSTs of the **P1**, **P2**, and **P3** copolymers were 30.0, 35.0, and 45.0 °C, respectively, as shown in Table 1 and Appendix A. The thermal decomposition thermograms of the prepared copolymers are shown in Appendix A. The thermal decomposition temperatures (*T*_d_) of **P1**, **P2**, and **P3** were 343, 351, and 352 °C, as shown in Table 1, which exhibited favorable and stable thermal properties.

### 3.2. Morphologies of Electrospun Nanofibers 

Figure 1a shows the field-emission scanning electron microscopy images of the as-spun ES nanofibers prepared from poly(NIPAAm-*co*-NMA-*co*-RhBN2AM) copolymers (**P1**, **P2**, and **P3**) in dry and wet states. These ES nanofibers were cross-linked by thermo-treating the samples at 120 °C for 48 h. The mechanism of NMA cross-linking is shown in Appendix A. The diameter ranges of the **P1**, **P2**, and **P3** ES nanofibers were estimated as 513 ± 28, 475 ± 35, and 412 ± 44 nm, respectively. The average diameter value was based on the statistical average of 50 fibers from each sample. Similar fiber diameters are a result of comparable polymer molecular weights, all approximately *M_n_* = 35,000 (Table 1) and the use of similar ES process parameters. Furthermore, toward ES nanofibers, the average diameter value before and after thermo-treatment was approximately equal (Appendix A). It means that thermal cross-linking does not affect the fiber morphology and diameter.

To observe the morphology of the cross-linked **P1**, **P2**, and **P3** ES nanofibers after Hg^2+^ or H^+^ sensing in an aqueous solution, the nanofibers were collected on a small piece of aluminum foil and immersed in water with Hg^2+^ or placed under an acidic condition. After detection, the samples were dried by freeze-drying for 30 min to remove the residual water to retain the original morphology. Figure 1a shows SEM images of the **P1**, **P2**, and **P3** ES nanofibers in the wet state after 10^−3^ M Hg^2+^ sensing. In comparison with the irregular films that derived from the **P1** and **P2** ES nanofibers, the **P3** ES nanofibers maintained a cylindrical shape after being extracted from the aqueous solution, as shown in Figure 1a (wet state, 20 °C), and exhibited fiber diameters of approximately 1 µm. The results indicated that **P1** was mostly soluble and **P2** was slightly soluble in water because of the insufficient NMA content for cross-linking to maintain the fiber morphologies. These swollen fibers maintained their cylindrical shape and did not dissolve in water because of the efficient chemical cross-linking of the NMA moieties, a result similar to that of our previous studies [28,29,31,35,37]. However, the fiber diameters of the **P2** and **P3** ES nanofibers shrank significantly (approximately 2.5-fold, from 1000 to 400 nm) after the nanofibers were heated in water (wet state, 60 °C; temperature above LCST) because the PNIPAAm chain (thermo-responsive) disintegrated in water, as shown in Figure 1a (wet state, 60 °C) and Figure 1b. The fiber diameters would vary as the temperature changed. In fact, the diameter decreased while temperature increased from 20 to 60 °C, effectively enhancing the sensitivity of the fluorogenic chemosensors, as demonstrated by PL spectra.

### 3.3. pH Sensing Property of ES Nanofibers

Figure 2a presents the PL spectra of **P2-fibers** in an aqueous solution with various pH levels (pH 7–2) at room temperature. The PL intensity of emission maxima (λ_PL, max_) at 584 nm gradually increased as the pH value decreased (from pH 7 to 2). When RhBN2AM detected H^+^, the RhBN2AM transformed from its non-fluorescent spirocyclic form into its opened cyclic form, a strong fluorescence, with λ_PL, max_ at 584 nm. As the inset of the figure shows, the PL intensity was quenched at pH 12, indicating that the poly(NIPAAm-*co*-NMA-*co*-RhBN2AM) copolymers were pH-dependent. In Figure 2b, the tendency of **P3-fibers** is identical to that shown in Figure 2a. In the inset of Figure 2b, **P3-fibers** exhibited on/off switchable fluorescence emission at pH 12 and 2. However, the λ_PL, max_ of **P3-fibers** was higher than that of **P2-fibers** because the cross-linked **P3-fibers** possessed a higher content of NMA. This would enable them to maintain their cylindrical shape and a higher surface-to-volume ratio. The results indicated that **P3-fibers** were more sensitive toward pH changes than **P2-fibers**. Figure 2c presents the thermo-dependent characteristics of **P2-fibers** at a constant pH level (pH 2). As the temperature increased from 27 to 55 °C, the PL intensity of the λ_PL, max_ at 584 nm increased (approximately 3.4-fold) for **P2-fibers** due to the disintegration of the thermo-responsive polymer chains (PNIPAAm) (55 °C; temperature above LCST). This is because the probe-to-probe (RhBN2AM to RhBN2AM) distance within the copolymer chains became narrower. The aggregation of probes led to the enhancement of the λ_PL, max_, and the results were similar to those reported in previous studies [10,34]. Despite thermo-dependent characteristics similar to those of **P2-fibers** (Figure 2d), the λ_PL, max_ of **P3-fibers** exhibited a 4.8-fold enhancement, indicating that **P3-fibers** were more dependent on pH and temperature than **P2-fibers**.

### 3.4. Hg^2+^ Sensing Property of ES Nanofibers

RhBN2AM is a favorable fluorescent probe for sensing pH and Hg^2+^. **P3-fibers** maintained their highly fibrous shape, which enabled excellent sensitivity. Thus, the capability of **P3-fibers** for sensing Hg^2+^ was explored. Figure 3a shows the PL spectra of **P3-fibers** at different Hg^2+^ concentrations. As the Hg^2+^ concentration increased, the λ_PL, max_ at 584 nm of **P3-fibers** gradually increased from 10^−7^ to 10^−3^ M because more Hg^2+^ ions were chelated by the RhBN2AM within the ES nanofibers, and the RhBN2AM transformed from its non-fluorescent spirocyclic form into its strongly fluorescent opened cyclic form, with λ_PL, max_ at 584 nm. In Figure 3b, **P3-film** exhibited similar fluorescent behavior to that of **P3-fibers**. Nevertheless, Figure 3c shows that **P3-fibers** exhibited an excellent capacity for Hg^2+^ sensing as compared with **P3-film**. This is because the surface-to-volume ratio of **P3-fibers** was higher than that of **P3-film**. These results were similar to those of our previous study [28].

Figure 4a shows the PL intensity variation of **P3-fibers** with the temperature at a constant Hg^2+^ concentration (10^−3^ M). The λ_PL, max_ at 584 nm of **P3-fibers** gradually increased (approximately 4.7-fold) when the temperature increased from 27 to 55 °C. Figure 4b shows the plot of intensity as a function of temperature for **P3-fibers**. Thermo-induced second fluorescence intensity enhancement was achieved at temperatures above the LCST (45 °C) because the area of the **P3-fibers** membrane decreased from 1.0 cm × 0.7 cm (specific size) to 0.8 cm × 0.5 cm, as shown in the figure inset, thereby exhibiting significant fluorescence emission after Hg^2+^ sensing under an ultraviolet (UV) lamp. The thermo-induced second fluorescence intensity enhancement was possibly due to the reduced fiber diameter, resulting in aggregated and dense Rh-chelated Hg^2+^ moieties.

Figure 5a shows the PL spectra recorded in the presence of Hg^2+^ or coexistence with other competing metal ions (Fe^2+^, Pb^2+^, Mg^2+^, Zn^2+^, Cu^2+^, and K^+^) in an aqueous solution. The black bars denote that the presence of Hg^2+^ induced the most significant PL intensity enhancement (approximately 31.7-fold) at 584 nm (λ_PL, max_), leading to red emission. The red bars denote that the PL spectra recorded in the coexistence of Hg^2+^ ions and other competing metal ions, revealing virtual unaffectedness for Hg^2+^ sensing. This indicates that **P3-fibers** are highly selective toward Hg^2+^ with almost no interference by these common coexisting of HTMs in the environment. Figure 5b shows that the fluorescence emission (λ_PL, max_ at 584 nm) of **P3-fibers** was observably enhanced after the completion of Hg^2+^-sensing. The fluorescence emission almost completely quenched by adding EDTA. Moreover, the on/off switchable fluorescence emission of **P3-fibers** in aqueous media that occurred upon the sequential addition of 10^−3^ M Hg^2+^ and EDTA could be repeated for at least five times. The spirocyclic form of RhBN2AM can be effectively induced to its opened cyclic form through binding with Hg^2+^ ions; this conversion was responsible for the on/off switchable fluorescence emission (the insets show the fluorescence emission variation of the **P3-fibers** membrane recorded under a UV lamp).

### 3.5. Thermo-Responsive Volume and Luminescence Variation of ES Nanofibers

**P3-fibers** exhibited an on/off switchable fluorescence emission in the confocal microscopy images as shown in Figure 6a. The strong and switchable fluorescence emission was due to the chelation of Hg^2+^ or protonation of H^+^ by RhBN2AM (all inset images were recorded under visible light). Figure 6b shows the on/off fluorescence emission and thermo-induced second fluorescence intensity enhancement of the **P3** ES nanofibrous membrane (1.0 cm × 0.7 cm) recorded under a UV lamp. Although the **P3** ES nanofibrous membrane was non-fluorescent in its original state (blank), red fluorescence emission could easily be activated by adding an acidic or Hg^2+^ aqueous solution. However, as shown in Figure 6b, the area of the ES fibrous membrane changed from 1.0 × 0.7 to 0.8 × 0.5 cm as the temperature increased from 25 to 60 °C. This indicates that the ES fibrous membrane exhibited a rapid and significant shrinkage by 57% of its initial area within 5 min. In addition, the PL intensity of the shrunk state at 60 °C was stronger than that of the swollen state at 25 °C under UV light illumination, as shown in the images of Figure 4b and Figure 6b, respectively. The thermo-induced second fluorescence intensity enhancement was possibly due to the reduced fiber diameter, which resulted in aggregated and dense Rh-chelated Hg^2+^ moieties. Figure 7 presents the schematic for preparing the sensory fibrous membranes (1 cm × 0.7 cm) with a porous architecture based on **P3** sensory copolymers, with the intention to apply them in the filtration of industrial wastewater containing specific HTMs, as well as environmental pH and temperature sensing. The robustness of the fibrous membranes with high absorption efficiency in Hg^2+^ filtration is similar to that in our previous research [28], as indicated by the significantly decreased concentration of Hg^2+^ ions in a very short period of time. The **P3**-based ES nanofibrous membranes with a porous architecture and high surface-to-volume ratios possess the considerable potential to be applied in water purification because of their effectiveness in absorbing Hg^2+^ in an aqueous environment. In addition, the fluorescence emission could be drastically enhanced at temperatures above the LCST. The rapid thermo-induced fluorescence emission or volume variation is due to the high surface-to-volume ratio of the ES nanofibers. Based on the above, the **P3**-based ES nanofibrous membranes have the potential to play the role of multifunctional sensory devices.

## 4. Conclusions

Novel multifunctional ES fibrous membranes with on/off switchable fluorescence emission properties, thermo-induced second fluorescence intensity enhancement characteristics, and high sensitivity toward pH and Hg^2+^ were prepared using poly(NIPAAm-*co*-NMA-*co*-RhBN2AM) (**P3**) by employing an electrospinning process. The NIPAAm, NMA, and RhBN2AM moieties were designed to provide hydrophilic and thermo-responsive properties, chemical cross-linking feature, and fluorescent probes, respectively. The **P3** ES nanofibers’ sufficient NMA content was capable of maintaining their fibrous cylindrical shape when immersed in the aqueous solution for pH and Hg^2+^ sensing. The fluorescence emission of the RhBN2AM within the ES nanofibers was highly sensitive toward pH and Hg^2+^ (i.e., non-fluorescence in neutral or alkaline media or an aqueous solution without Hg^2+^ (spirocyclic form), but strong fluorescence in acidic media or aqueous solutions with Hg^2+^ (ring-opened cyclic form)). Thus, the on/off switchable properties of the fluorescence emission can be easily modulated by tuning the pH and Hg^2+^ concentration. In addition, the degree of disintegration of the **P3** ES nanofibers was dependent on the temperature, resulting in a decrease in the probe-to-probe (RhBN2AM–RhBN2AM) distance and further enhancement of the red fluorescence emission. Substantial reversible PL emission from the **P3** ES nanofibrous membrane was also observed in Hg^2+^ and EDTA aqueous solutions for at least five times. This study demonstrates that the prepared fluorescent ES nanofibrous membranes, which can be used as “naked eye” sensors, have marked advantages for applications in multifunctional devices for specific HTM chelation, as well as pH and temperature sensing in aqueous environments.

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
