# Peer review of "Novel Multifunctional Luminescent Electrospun Fluorescent Nanofiber Chemosensor-Filters and Their Versatile Sensing of pH, Temperature, and Metal Ions"

_polymers, 2018, doi:10.3390/polym10111259_

Round 1

Reviewer 1 Report

The present job is a further study investigation of the authors previous publication in Dyes Pigm 2017, 143, 129. Although the method is a simple one-fluid blending electrospinning process, the applications are interesting and should draw broad attention from the readers. The experiments were well arranged and the manuscript is well in preparation. I recommend its acceptance for publication after the following issues are well addressed.

1. It is better to provide some quantative results in the ABSTRACT.

2. In the second paragraph, a state-of-art conclusions about the development of electrospinning should be provided, some recent important publications should be cited for provoking the readers’ thoughts on the further investigations along this job, such as coaxial electrospinning (put the functional molecules only on the surface), side-by-side electrospinning, and other types of multiple-fluid electrospinning (Eur J Pharm Sci 122 (2018) 195-204//J Control Release, 2018, Doi:10.1016/j.jconrel.2018.08.016//Chem Eng J 356(2019) 886-694// Carbohydr Polym 2019, 203, 228-237// Nanomaterials 2018, 8, 184). these new technologies can provide better tools for the next study along your job.

3. Often in one manuscript, it is better to use phrase such as “in this study” / ”in the present study” in manuscript only one time for the most important contents.

4. How to control the thicknesses of your casting films? The preparation should have more details for the readers to duplicate.

5. As a fast developing field, only 7 from 40 of the reerences is within the most recent three years. The common standard is 30%.     

Author Response

 We appreciate the comments from the reviewers. The following is our point-to-point response to the comments.

Review-1

The present job is a further study investigation of the authors previous publication in Dyes Pigm 2017, 143, 129. Although the method is a simple one-fluid blending electrospinning process, the applications are interesting and should draw broad attention from the readers. The experiments were well arranged and the manuscript is well in preparation. I recommend its acceptance for publication after the following issues are well addressed.

1. It is better to provide some quantative results in the ABSTRACT.

Ans.: Thank you for the comment. We has added the following description, some quantative results, to ABSTRACT of the revised manuscript: (the λPL, max exhibited a 4.8-fold enhancement) or Hg2+ (at a constant Hg2+ concentration (10-3M), the λPL, max of P3-fibers exhibited 4.7-fold enhancement), and high reversibility of on/off switchable fluorescence emission at least five times”.

2. In the second paragraph, a state-of-art conclusions about the development of electrospinning should be provided, some recent important publications should be cited for provoking the readers’ thoughts on the further investigations along this job, such as coaxial electrospinning (put the functional molecules only on the surface), side-by-side electrospinning, and other types of multiple-fluid electrospinning (Eur J Pharm Sci 122 (2018) 195-204//J Control Release, 2018, Doi:10.1016/j.jconrel.2018.08.016//Chem Eng J 356(2019) 886-694// Carbohydr Polym 2019, 203, 228-237// Nanomaterials 2018, 8, 184). these new technologies can provide better tools for the next study along your job.

Ans.: Thank you for the comment, some of your suggested references have been cited. We added these references and the following description to page 2 of the revised manuscript:such as coaxial electrospinning [23, 24], side-by-side electrospinning [25], and other types of multiple-fluid electrospinning [26, 27].”.

3. Often in one manuscript, it is better to use phrase such as “in this study” / ”in the present study” in manuscript only one time for the most important contents.

Ans.: Thank you for the comment. We deleted some phrase such as “in this study” / ”in the present study” in manuscript. Thus, we just use only one time for the most important contents in the page 2 of the revised manuscript.

4. How to control the thicknesses of your casting films? The preparation should have more details for the readers to duplicate.

Ans.: Thank you for the comment. We used the same concentration and volume of polymeric solution to prepare nanofibers and thin film, respectively. Because we used the same conditions for every thin film-type samples, their thickness of all samples is controlled at 18 μm. Furthermore, we provided their preparations in 2.5. Preparation of electrospun fibers and drop-cast films. “On the other hand, the corresponding films of P1,P2, and P3 were respectively drop-cast on glass substrates using the same concentration and volume of the polymer solution.”

5. As a fast developing field, only 7 from 40 of the references is within the most recent three years. The common standard is 30%.

Ans.: Thank you for the comment. We deleted 5 older references and further added 5 references you recommended. After checking, 12 form 40 of the references is within the most recent three years. The common standard reached to be at 30%.

Reviewer 2 Report

In this manuscript, the authors reported an interesting work about multifunctional electrospun fibrous fluorescent chemosensors with high sensitivity toward pH, mercury ions (Hg2+), and temperature. The experiments are well designed, and the characterizations are comprehensive. Therefore, it is commended for publication in Polymers after minor revision.

Please find the following commons for the authors’ reference.

1. In the introduction, please further explain the novelty/significance of combining temperature, pH and Hg2+ sensing.

2. Please explain how the thermal crosslinking parameters (120 ºC and 48h) are determined?

3. Does the thermal crosslinking affect the fiber morphology/diameter? It is suggested to compare the SEM images and diameter changes before and after the thermal crosslinking.

Author Response

We appreciate the comments from the reviewers. The following is our point-to-point response to the comments.

Review-2

In this manuscript, the authors reported an interesting work about multifunctional electrospun fibrous fluorescent chemosensors with high sensitivity toward pH, mercury ions (Hg2+), and temperature. The experiments are well designed, and the characterizations are comprehensive. Therefore, it is commended for publication in Polymersafter minor revision.

Please find the following commons for the authors’ reference.

1. In the introduction, please further explain the novelty/significance of combining temperature, pH and Hg2+ sensing.

Ans.:Thank you for the comment. We addedthe following description to page 3 of the revised manuscript:Furthermore, the fluorescence emission of the novel chemosensors exhibited second enhancement above the LCST, it indicated the chemosensors could be applied in pH (or Hg2+) and temperature simultaneously.”. It means that our fibrous fluorescent chemosensors can directly exhibit temperature sensing even after they detect pH and Hg2+. This is the novelty/significance point.

2. Please explain how the thermal crosslinking parameters (120 ºC and 48h) are determined?

Ans.:Thank you for the comment. The NMA moiety is a chemical cross-linkable segment. Their intermolecular crosslinking reaction has been shown in Scheme S1 ofSupporting Information (pages 2). We further added the following description to page 6 of the revised manuscript:The mechanism of NMA crosslinking is shown in Scheme S1“. Therefore,P1ES nanofibers were dissolved by the aqueous solution and form an irregular film. This indicates that the P1 was mostly soluble in water because of the insufficient NMA content for crosslinking to maintain the fiber morphology. P2 ES nanofibers in wet state, the nanofibers were slightly destruction. On the contrary, P3 ES nanofibers were relatively stable in aqueous solution and maintained better cylindrical shape without dissolving in water, attributed to the relatively higher content of chemically cross-linkable NMA moieties.

3. Does the thermal crosslinking affect the fiber morphology/diameter? It is suggested to compare the SEM images and diameter changes before and after the thermal crosslinking.

Ans.:Thank you for the comment. After and before thermo-treatment, the average diameters of all ES nanofibers are similar. It means that thermal crosslinking does not affect the fiber morphology/diameter. These results had been shown in Figure S6 of Supporting Information (page 4). We also added following description to page 6 of the revised manuscript:Furthermore, toward ES nanofibers, the average diameter value before and after thermo-treatment was approximately equal (Figure S6). It means that thermal crosslinking does not affect the fiber morphology and diameter.”

Reviewer 3 Report

Chen et al. proposed a fluorescent nanofiber chemiosensors (P3 ES) and their potential possibility for the industrial waste water treatment. The composition of P3 ES (NIPAAm: NMA: RhB = 71.1: 28.4: 0.5) was characterized by GPC, and 1H NMR. High temperature decreased the diameter of P3 ES, leading to increasing the fluorescence. In addition, Hg(II) ions were chelated by the RhBN2AM, as a result of transformed from its non-fluorescent spirocyclic form into its strongly fluorescent opened cyclic form. Thus P3 ES provided higher selectivity for Hg(II). Overall, this study is interesting and the manuscript is written well. However, some unclear parts are found and should be provided response before accepted for publication.

Figure 2-5, no suitable label (number) is found in y axis. It is difficult to compare the fluorescent “turn on” effect when different temperature, pH value and Hg(II) concentration. As author mentioned, when RhBN2AM is presence with H+, it transformed form its non-fluorescent spirocyclic form into its opened cyclic form (LL232-234). The same reason was provided when Hg(II) added into the P3 ES (LL262-234). Which one make P3 ES stronger fluorescence?

In your research design, what is the stoichiometry between Hg(II) to P3 ES? What is the maximum capacity for the Hg(II) ions detection?

How about the tolerance when Hg(II) and other metal ions were coexisted?

Whether or not the proposed P3 ES fibers (films) produced “acid error (like pH glass electrode” when sensing Hg(II) ions?

How about the reaction time make P3 ES fluorescence change?

If reaction time was longer than 20 min, is it possible to use this P3 ES for the filtration of industrial wastewater?

Author Response

We appreciate the comments from the reviewers. The following is our point-to-point response to the comments.

Review-3

Chen et al. proposed a fluorescent nanofiber chemiosensors (P3 ES) and their potential possibility for the industrial waste water treatment. The composition of P3 ES (NIPAAm: NMA: RhB = 71.1: 28.4: 0.5) was characterized by GPC, and 1H NMR. High temperature decreased the diameter of P3 ES, leading to increasing the fluorescence. In addition, Hg(II) ions were chelated by the RhBN2AM, as a result of transformed from its non-fluorescent spirocyclic form into its strongly fluorescent opened cyclic form. Thus P3 ES provided higher selectivity for Hg(II). Overall, this study is interesting and the manuscript is written well. However, some unclear parts are found and should be provided response before accepted for publication.

1. Figure 2-5, no suitable label (number) is found in y axis. It is difficult to compare the fluorescent “turn on” effect when different temperature, pH value and Hg(II) concentration. As author mentioned, when RhBN2AM is presence with H+, it transformed form its non-fluorescent spirocyclic form into its opened cyclic form (LL232-234). The same reason was provided when Hg(II) added into the P3 ES (LL262-234). Which one make P3 ES stronger fluorescence?

 Ans.:Thank you for the comment. The label (number) of Figure 2-5 has been attached on pages 8-12. However, the intensity (y-axis) of Figure 3(c) and Figure 5 is normalized. In this study, pH 2 (acidic consition) added into the P3 ES fibers make them stronger fluorescence than Hg(II) addition.

2. In your research design, what is the stoichiometry between Hg(II) to P3 ES? What is the maximum capacity for the Hg(II) ions detection?

Ans.:Thank you for the comment. (1) According to the reference 7 (Dalton Trans. 2011, 40, 12578-83) and our research design, it shows the good agreement of stoichiometry between Hg(II) and sensory moiety (RhBN2AM) is 2:1. (2) An absorption capacity could achieve 80.56%, as indicated by the concentration of Hg2+ ions strongly decreasing from 399.49 to 77.65 ppm in 10 min. The results are similar to our previous report (Dyes Pigm. 2017, 143, 129-142.).

3. How about the tolerance when Hg(II) and other metal ions were coexisted?

Ans.:In Figure 5(a), P3 ES nanofibers exhibited strong emission in an environment with Hg2+ or Hg2+ and other metal ions coexisted. This indicates that the coexistent metal ions does not affect the selectivity of P3 ES nanofibers.

4. Whether or not the proposed P3 ES fibers (films) produced “acid error (like pH glass electrode” when sensing Hg(II) ions?

Ans.:Thank you for the comment. To avoid acid error situation, the Hg2+ aqueous solution (controlled at pH=7) was prepared by diluting Hg2+ stock solution (0.1 M) with deionized water and Tris-HCl buffer solution.

5. How about the reaction time make P3 ES fluorescence change?

Ans.:Thank you for the comment. The emission of probes (RhBN2AM) within hydrophilic copolymer can be induced by acid solution or chelating Hg2+. Therefore, the fluorescence change of P3 ES nanofibers is depending on the absorption of Hg2+ in aqueous solution. In our study, for the reaction time of P3 ES nanofibers, the color change can be observed at 10-3 M Hg2+ aqueous solution (~6min), but at pH=2 aqueous solution (~2min) by “naked eye”.

6. If reaction time was longer than 20 min, is it possible to use this P3 ES for the filtration of industrial wastewater?

Ans.:Thank you for the comment. The measurement of P3ES nanofibrous membrane (1 cm × 0.7 cm) chelated Hg2+ (3 mL) in an aqueous solution at a concentration of 399.49 ppm by using ICP-mass. An absorption capacity could achieve 80.56%, as indicated by the concentration of Hg2+ions strongly decreasing from 399.49 to 77.65 ppm within 10 min. The results are similar to our previous report (Dyes Pigm. 2017, 143, 129-142.). Thus, the reaction time was smaller than 10 min, we think that it is potential to use this P3 ES for the filtration of industrial wastewater in the future.

Round 2

Reviewer 1 Report

The manuscript's quality has been substantially improved. I recommend its acceptance for publication in POLYMERS in its present form.

Reviewer 3 Report

The questions are provided good response. The text could be accepted in current form.